# Effects of Ultrasound Combined with Preheating Treatment to Improve the Thermal Stability of Coconut Milk by Modifying the Physicochemical Properties of Coconut Protein

**DOI:** 10.3390/foods11071042

**Published:** 2022-04-04

**Authors:** Yizhou Sun, Haiming Chen, Wenxue Chen, Qiuping Zhong, Ming Zhang, Yan Shen

**Affiliations:** 1College of Food Science and Engineering, Hainan University, 58 Renmin Road, Haikou 570228, China; 19085231210026@hainanu.edu.cn (Y.S.); hmchen168@163.com (H.C.); hnchwx@vip.163.com (W.C.); hainufood88@163.com (Q.Z.); zhangming-1223@163.com (M.Z.); 2Maritime Academy, Hainan Vocational University of Science and Technology, 18 Qiongshan Road, Haikou 571126, China; 3College of Tropical Crops, Hainan University, 58 Renmin Road, Haikou 570228, China

**Keywords:** preheating, ultrasound, coconut globulin, coconut milk

## Abstract

In the food industry, coconut milk has a unique flavor and rich nutritional value. However, the poor emulsifying properties of coconut proteins restrict its development. In this study, the effect of ultrasound combined with preheating on coconut globulin and coconut milk was evaluated by physicochemical properties and structural characteristics. The results showed that ultrasound and 90 °C preheating gave coconut protein better emulsifying and thermal properties, demonstrated by higher solubility (45.2% to 53.5%), fewer free sulfhydryl groups (33.24 to 28.05 μmol/g) and higher surface hydrophobicity (7658.6 to 10,815.1). Additionally, Fourier transform infrared spectroscopy and scanning electron microscopy showed obvious changes in the secondary structure. Furthermore, the change in the physicochemical properties of the protein brought a higher zeta potential (−11 to −23 mV), decreased the thermal aggregation rate (148.5% to 13.4%) and increased the viscosity (126.9 to 1103.0 m·Pa·s) of the coconut milk, which indicates that ultrasound combined with preheating treatment provided coconut milk with better thermal stability. In conclusion, ultrasound combined with preheating will have a better influence on modifying coconut globulin and increasing the thermal stability of coconut milk. This study provides evidence that ultrasound and other modification technologies can be combined to solve the problems encountered in the processing of coconut protein products.

## 1. Introduction

Coconut milk is one of the most important plant protein drinks, and it has a unique flavor and rich nutritional value [1]. One study indicated that the global coconut milk market was estimated to be USD 814.4 million in 2020 and that it would grow at a 9.3% compound annual growth rate to reach USD 1775.4 million by 2027 [2]. Another study showed [3] that coconut milk was a good substitute for cow milk. On the one hand, cow milk contains a large amount of lactose, which has the highest allergenic potential, while coconut does not contain lactose [3]. On the other hand, the fat content of cow milk is 3.27 g/100 mL, and for coconut milk it is 5.04 g/100 mL [4]. However, approximately 87% of the fat from coconut milk is lauric acid, followed by caprylic acid and decanoic acid (13%). This fat is more easily metabolized by the body, and the energy is only 70 kcal/100 mL, which is lower than for cow milk (150 kcal/100 mL) [3]. At the same time, this fat is beneficial to prevent arteriosclerosis and related diseases. Compared with soy milk, the protein content of coconut milk (1.28 g/100 mL) is lower than for soy milk (2.88 g/100 mL) [4]. Yet, soy protein is considered an allergen, and soy milk carries a higher risk of allergy [5]. Coconut protein does not have the risk of allergy, and it contains a large number of essential amino acids (71–77%), which is more easily digested and absorbed by the body [6]. Moreover, compared with soy milk, coconut milk contains more minerals (particularly calcium, phosphorus and potassium) and vitamins (i.e., vitamins C, E, B1, B3, B5 and B6), which have strong antioxidant activity [3]. Therefore, coconut milk is an excellent vegetable protein drink.

However, the stability of coconut milk is not easily maintained due to the limited emulsifying properties of coconut protein, especially globulin [7]. The granular sensation and sedimentation of coconut milk during processing and storage have a negative impact on the sensory properties and flavor [8,9,10]. Moreover, the heating treatment in high-pressure steam sterilization and ultra-high-temperature instantaneous sterilization can cause denaturation of coconut globulin. This will induce the exposure of sulfhydryl groups and hydrophobic groups to form protein aggregation and gelation, which will induce phase separation and flocculation of coconut milk [11]. Therefore, it is a challenge to maintain the emulsion’s stability by changing the physicochemical properties of protein during sterilization.

Ultrasound, as an emerging physical method for modifying the structure of proteins, has attracted extensive attention [12]. The energy generated by ultrasound mainly comes from cavitation. Cavitation can lead to the formation of small, rapidly growing bubbles in a fluid medium, and these bubbles will implode violently, finally, to generate high temperatures, pressures, and shear forces, which will release a large amount of energy [13,14]. Therefore, ultrasound can induce changes in protein conformation and affect its physicochemical properties [15]. Previous studies have shown that ultrasound can accelerate protein unfolding and lead to protein denaturation. Therefore, protein will expose a large number of hydrophobic groups and form many active sites on the surface, which is considered to change the structure and improve the functional characteristics of protein [16]. However, as a non-thermal green technology, the modification capabilities of ultrasound is insufficient, and changes in protein structure cannot fully meet the requirements of the food industry [17]. Therefore, the combination of ultrasound and other protein modification technologies has become the focus of studies in the food industry. Preheating is a simple and convenient technique to modify the functional properties of protein such as solubility, emulsibility, and foamability [18]. During preheating, high temperatures will cause protein denaturation, which can result in the exposure of some groups buried inside protein molecules and an increase in the surface charge of the protein. At the same time, denaturation will cause quenching of the active site of the protein and then induce the recombination of intramolecular and intermolecular bonds in the protein [19].

The coconut milk industry plays an important role in the economic development of tropical areas. Coconut milk is high in fat, stabilized by natural coconut protein and has excellent nutritional value. Due to the poor emulsifying properties of coconut protein, coconut milk is prone to becoming unstable and stratified during the production, transportation and storage process, which affects the product’s quality. Therefore, it is important to maintain the stability of coconut milk in the industry, studies on improving the protein’s emulsifying properties by preheating and ultrasound can reduce the use of additives and ensure the safety, nature and health of the products. At the same time, it is beneficial to reduce the production cost, optimize the production process and promote the development of the coconut industry. However, previous studies have mainly focused on the relationship between temperature and heating time. Only a few studies have shown that high temperature combined with high-pressure technology can improve emulsion stability by changing the protein’s structure and producing controllable protein particles [20,21]. However, the effect of the combination of ultrasound and preheating is still a blank field. Thus, as inspired, in this study, we present a simple yet robust pathway for modifying coconut protein using ultrasound combined with preheating. In this study, ultrasound was subjected to a preheating treatment to explore their synergistic effects. The changes in the structural characteristics and physicochemical properties of coconut globulin were examined by solubility, surface hydrophobicity, free sulfhydryl group (SH), Fourier transform infrared (FTIR) spectroscopy and scanning electron microscopy. The stability of the coconut milk was investigated by zeta potential, particle size, particle size distribution and rheological properties. The study of the synergistic effect of ultrasound and preheating on coconut globulin and coconut milk can be used to improve the production, processing and storage stability and for extending the shelf life of products.

## 2. Materials and Methods

### 2.1. Materials

Fresh coconut milk was purchased from Taifengyuan Food Company in Haikou, China. 1-Aniline naphthalene-8-nitrobenzoate (ANS) and 5,5’-dithiobis-(2-nitrobenzoic acid) (DTNB) were purchased from Sigma Chemical Company (St. Louis, MO, USA). All other reagents used were of analytical grade.

### 2.2. Methods

#### 2.2.1. Extraction of Coconut Globulin

Coconut globulin was extracted by modified Osboren method [22]. Coconut globulin was isolated from frozen coconut milk using 0.5 mM NaCl solution. The sample and NaCl solution were mixed with 1:10 (*w*/*v*), and the mixture was stirred by magnetic force at 4 °C for 15 h. The insoluble residue was removed by centrifuging at 10,000 r/min for 30 min at 4 °C. The extraction was repeated three times, and the supernatant obtained from each extraction was combined and diluted into a 20-fold volume by deionized water. Finally, the solution was subjected to freeze drying, and the coconut globulin dry powder obtained was stored at −20 °C until further analysis.

#### 2.2.2. Modification of Coconut Globulin and Coconut Milk

The coconut globulin (0.1% *w*/*v*) was dissolved in phosphate buffer (10 mM, pH 7.2) and stirred thoroughly. Then, the solution was divided into two groups. The first group was heated in a HH-4 water bath (Changzhou Aohua Inetrument Co., Ltd., Guangzhou, China) at 70, 80 or 90 °C for 20 min. The second group was heated and treated to ultrasound (40 *w*/*l*, 53 kHz) simultaneously at 70, 80, or 90 °C for 20 min using a XC-3000 ultrasonicator (Jining Xinxin ultrasonic electronic equipment Co., Ltd., Jining, China). After homogenizing, the samples were heated at 135 °C for 60 s in a DF-101S water bath (Shanghai Yushen Inetrument Co., Ltd., Shanghai, China). Finally, all the samples were freeze-dried. The powders were obtained and stored at −20 °C before further use.

#### 2.2.3. Solubility of Coconut Protein

Coconut globulin powder was dissolved in phosphate buffer (10 mM, pH 7.2) to reach the condition 0.1% *w*/*v* and stayed overnight at 4 °C for full hydration. Then, the solution was centrifuged at 10,000 r/min for 30 min and filtered through filter paper. The supernatant was divided into six groups according to the method described in Section 2.2.2., and the solubility was determined by the Coomassie brilliant blue method [23]. The solubility was calculated using the following equation:Solubility = 100 (P_S_/P_T_) 
where P_S_ (mg/mL) is the protein concentration remaining in the supernatant after centrifugation and filtration, and P_T_ (mg/mL) is the total protein concentration present in the original solution.

#### 2.2.4. Surface Hydrophobicity of Coconut Protein

The surface hydrophobicity was determined according to methods of Kato [24] and Haskard [25] with slight modification. The globulin powder was dissolved in the above phosphate buffer to obtain several concentrations within the range of 0.005–0.500 mg/mL. Then, 40 μL of ANS solution (10 mM in the above phosphate buffer) was added to 3 mL of globulin solution and mixed immediately. The fluorescence intensity was determined using an F-7000 spectrophotometer (Hitachi, Tokyo, Japan) at an excitation wavelength of 370 nm, emission wavelength of 490 nm and slit of 5 nm. The protein concentration was the abscissa, the fluorescence intensity was plotted as the ordinate, and the slope of the linear regression was the surface hydrophobicity.

#### 2.2.5. SH of Coconut Protein

The SH of globulin was determined according to method of Beveridge et al. [26]. Two mL of globulin solution was diluted into 10 mL by Tris-Gly buffer (0.086 mol/L Tris, 0.090 mol/L Gly, 0.004 mol/L EDTA, 8 mol/L urea, pH 8) and mixed immediately. Then, 80 μL of DTNB (4 mg/mL) was added to protein samples to react for 15 min at room temperature. Afterwards, the absorbance value was measured at a wavelength of 412 nm. The calculation formula for SH is as follows:SH (μmol/g protein) = 73.53A_412_D/C 
where 73.53 = 106/(1.36 × 104), 1.36 × 104 is the molar extinction coefficient of Ellman’s reagent, A_412_ is the differentials absorbance at 412 nm, D is dilution factor and C is the protein concentration (mg/mL).

#### 2.2.6. FTIR Spectroscopy of Coconut Protein

The FTIR spectra of coconut globulin were recorded by a T27 fluorophotometer (Bruker, Karlsruhe, Germany). The coconut globulin was mixed with KBr at a ratio of 1:100 and pressed into tablets for further measurement. The FTIR spectra were measured in the absorbance mode between 400 and 4000 cm^−1^, and the automatic signals gained were collected over 32 scans at a resolution of 4 cm^−1^ against a background spectrum recorded from the KBr pellet [27]. The secondary structure of the protein was analyzed in the range of 1600–1700 cm^−1^ by Peak Fit Version 4.12 (Systat Software Inc., San Jose, CA, USA) [28].

#### 2.2.7. Scanning Electron Microscopy of Coconut Protein

The microstructure of globulin was determined using a Verios G4 UC field emission scanning electron microscope (Thermo Scientific, Waltham, MA, USA) under an acceleration voltage of 5 kV to observe the samples’ surface morphology. Before determinizing, the samples were coated with double-sided conductive adhesive and gold.

#### 2.2.8. Zeta Potential of Coconut Milk

Coconut milk were diluted 100 times with deionized water and then injected into a static light scattering with a zetasizer nano ZS90 (Malvern Instrument, Malvern Hills, UK) [29]. The temperature was kept at 25 °C with a temperature controller. The zeta potential of the samples was measured three times.

#### 2.2.9. Particle Size and Size Distribution of Coconut Milk

The droplet size distribution and average particle size of the different coconut milk samples were measured by static light scattering with a Malvern Master Sizer 2000 instrument (Malvern Instruments, Malvern Hills, UK) [30]. Water was used as a dispersant. The refractive indices of the dispersed phase (coconut oil) and the dispersant medium (distilled water) were 1.456 and 1.330, respectively. The absorbance of the emulsion droplet was fixed at 0.001.

#### 2.2.10. Rheological Properties of Coconut Milk

The Haake Mars 40 rheometer (ThermoFisher, Waltham, MA, USA) was used to measure the rheological properties of coconut milk [31]. The plate diameter was 40 mm, and the slit distance was set as 1 mm. The samples around the plate were scraped with the weighing spoon. The variation in the apparent viscosity was measured at 25 °C, with a shear rate in the range of 0.1–50 s^−1^.

#### 2.2.11. Statistical Analysis

All experiments were carried out at least in triplicate. The results are shown as the mean ± standard deviation. The experimental data were plotted by Origin 2019 (Origin Lab Corporation, Northampton, MA, USA) and analysis by DPS software V18.10 (Zhejiang University, Hangzhou, China). The one-way ANOVA test (Tukey’s test) was used to determine the significant statistical differences. A *p*-value < 0.05 was considered as a significant difference.

## 3. Results and Discussion

### 3.1. Physicochemical Properties of Coconut Protein

#### 3.1.1. Solubility

Solubility is one of the most practical indexes that is closely related to the functional properties of proteins [32]. The solubility of coconut globulin treated by ultrasound, preheating and preheating combined with ultrasound is shown in Figure 1. For the untreated coconut globulin (control), the solubility was 69.2%, which was higher than the others. At the same time, the increase in the preheating temperature decreased the solubility of coconut globulin at 70 (58.0%), 80 (51.8%) and 90 °C (45.2%) conditions and presented a downward trend. During the preheating process, with the gradual increase in temperature, the degree of coconut globulin molecules unfolding increased and the hydrophobic amino acid groups buried in the molecules were exposed, which induced the hydrophobic aggregation of coconut globulin molecules and, ultimately, led to the decrease in coconut globulin solubility.

It is worth noting that when coconut globulin was modified by preheating and ultrasound together, the solubility of coconut globulin increased to 60.0% (70 °C-ultrasound), 54.7% (80 °C-ultrasound) and 53.5% (90 °C-ultrasound) compared with the preheating groups. The increase in solubility may be related to the cavitation of ultrasound. The high-speed shear force generated by ultrasonic cavitation can promote the unfolding of coconut globulin and destroy the non-covalent interactions among coconut globulin such as hydrogen bonds and hydrophobic interactions [32]. A previous study held that the increase in solubility might be due to the change in structural conformation. The energy generated by ultrasound would make native protein complexes separate into individual subunits [33]. At the same time, cavitation will also expose some hydrophilic groups of the protein, which will increase the hydration between protein molecules and water molecules. Therefore, the protein will show a relatively high solubility after ultrasound.

After high-temperature sterilization, the solubility of coconut protein showed a decreasing trend in general. This may be because high temperatures change the charge state of the coconut globulin’s surface. A high temperature will induce coconut globulin denaturation, which leads to rearrangement of the distribution of charged amino acids on the molecular surface. Therefore, the hydrophilic and hydrophobic ratios of coconut globulin will be unbalanced, and protein aggregation will form via hydrophobic interaction, which will decrease the solubility of coconut globulin. However, it is interesting to note that the solubility decrease rate of 90 °C-ultra group was only 8.5% (53.5% to 48.9%), while the decrease rate of the control group was 59.9% (69.6% to 27.9%). This was because the protein formed more soluble aggregations and had a much greater surface charge during the 90 °C preheating and ultrasound, which can help coconut globulin better resist high-temperature damage and maintain structural stability during the subsequent high-temperature sterilization.

#### 3.1.2. Surface Hydrophobicity

The surface hydrophobicity of protein can be expressed by the content of hydrophobic amino acid residues after protein side-chain exposure. Therefore, it was importantly correlated with protein emulsification [34]. Additionally, it was also significant to promote proteins to adsorb onto the surface of oil droplets and prevent protein coalescence and flocculation [35]. In Figure 2, the surface hydrophobicity of preheating groups had a significant increase compared with the control group. Especially, the surface hydrophobicity of the 90 °C group (7658.6) was more double times higher than the control group (3018.8). During the heating process, coconut globulin will unfold and expose many hydrophobic groups that were inside the molecule, thus increasing the surface hydrophobicity. This transferring of hydrophobic groups from inside to the molecular surface will be strengthened with the increase in temperature. When samples were treated by ultrasound combined with preheating, the surface hydrophobicity was further improved from 3741.6 (70 °C), 5781.9 (80 °C), 7658.6 (90 °C) to 4557.0 (70 °C-ultrasound), 8547.4 (80 °C-ultrasound), 10,815.1 (90 °C-ultrasound), respectively. The ultrasonic cavitation could generate a high-intensity shock wave, shear force and turbulence, which could have a synergistic effect with preheating to further promote the unfolding of coconut globulin and expose more hydrophobic groups, which would increase the hydrophobic activity.

Interestingly, in the subsequent high-temperature sterilization process, the increasing trend in surface hydrophobicity for the 90 °C-ultra group was significantly suppressed, and the increase rate was only 12.9%, while the increase rate for the control group was up to 44.4%. During preheating and ultrasound, coconut globulin exposed many hydrophobic groups and formed more suitable space conformations that could resist high temperatures. Thus, during high-temperature sterilization, the exposed hydrophobic groups could inhibit the intermolecular interactions effectively to maintain the structural stability of the protein. Therefore, the change rate of the surface hydrophobicity was smaller than the control group.

#### 3.1.3. SH

SH is an important active group in protein molecules that can significantly affect the functional properties of protein. This is because the interchange reaction of sulfhydryl and disulfide can promote the polymerization of proteins. This form of polymerization not only can causes the protein to adsorb onto the interface irreversibly, but it also provides a highly viscoelastic film to resist protein aggregation [36]. It can be seen from Figure 3 that the content of SH increased with the improved preheating temperatures. The SH content of coconut globulin was 22.75 μmol/g (control), 26.74 μmol/g (70 °C), 31.37 μmol/g (80 °C) and 33.24 μmol/g (90 °C). Under the stimulation of high temperature, the denaturation of coconut globulin would break the disulfide bond (S–S) and form SH. At the same time, the tendency of SH to form S–S was inhibited, thereby increasing the content of SH [37]. It is worth noting that the preheating combined with ultrasound decreased the SH content of coconut globulin at 70 °C-ultrasound (from 26.74 to 25.83 μmol/g), 80 °C-ultrasound (from 31.37 to 26.64 μmol/g) and 90 °C-ultrasound (from 33.24 to 28.05 μmol/g). The high temperature and high-pressure area caused by ultrasonic cavitation could hydrolyze water into hydrogen atoms and highly active hydroxyl radicals, which can promote the oxidation of SH to form S–S [38,39]. In addition, ultrasound can strengthen the Brownian motion between molecules, thereby increasing the probability of SH colliding to form S–S. Therefore, these two reasons together reduced the content of SH in the coconut globulin.

Interestingly, during the subsequent high-temperature sterilization process, the increase rate of SH in the control group reached 144.4%, while the increase rates in the preheating and ultrasound groups were only 36.0% (70 °C-ultrasound), 29.7% (80 °C-ultrasound) and 8.1% (90 °C-ultrasound), which were lower than the control group. Under the stimulation of high temperature, the coconut globulin would unfold, exposing a large amount of SH, thus causing a sharp increase in the content of SH in the control group. However, when the coconut globulin was treated by preheating and ultrasound together, the exposed hydrophobic groups would form a more stable spatial structure and better interface properties, which could improve the structural stability. Therefore, in the subsequent high-temperature sterilization process, the coconut globulin tended to undergo strong hydrophobic aggregation to wrap the SH by hydrophobic interactions. At the same time, SH cannot interact with oxygen or other SH, which prevented the sulfhydryl oxidation or SH–SS exchange reaction. Thus, the increase rate of SH was significantly slowed down, and it maintained the stability of coconut globulin during high-temperature processing.

#### 3.1.4. FTIR and Secondary Structure of Coconut Protein

FTIR can implicate the changes in the protein’s structure [40]. Previous studies have shown that FTIR spectra can present the vibrations of the amide bonds of proteins; therefore, it could be used to represent the secondary structure of proteins [41]. For proteins, the spatial structure was mainly expressed at the amide I band (within a wavelength of 1600–1700 cm^−1^), the amide II band (within a wavelength of 1530–1550 cm^−1^) and the amide III band (within a wavelength of 1260–1300 cm^−1^). In addition, a study has suggested that the amide I band was the closest relation to the protein’s secondary structure compared with the amide II and III bands.

It can be seen from Figure 4 that the peak position shifted significantly to the right and the intensity of the peak was enhanced. This is to say that the coconut globulin’s structure changed significantly after preheating and ultrasound. Furthermore, at the amide I band, the peak shifted from 1645 cm^−1^ (control) to 1639 cm^−1^ (70 °C), 1637 cm^−1^ (80 °C) and 1643 cm^−1^ (90 °C), and the intensity of the amide I band increased, which indicates an increase in the content of the α-helix and random coil. At a wavelength of 3500 cm^−1^, which was mainly reflected in the tensile vibration of OH or NH, the increase in the absorption peak’s intensity might mainly have come from the preheating, which exposed some free radical groups, such as OH or NH, in the protein structure. In addition, the free radical groups could form a hydrogen bond with the carbonyl group in the protein peptide chain. At the same time, the exposure of hydrophobic groups was also conducive to the attraction of O–H or N–H groups to form hydrogen bonds. When coconut globulin was treated by ultrasound and preheating, the peak of the amide I band had a slight left shift trend compared to the samples treated by preheating alone. This was possibly because the cavitation generated by ultrasound destroyed the hydrogen bonds that could maintain the stability of the α-helix. In addition, the decrease in the absorption peak’s intensity at 3500 cm^−1^, compared to the preheating alone, also indicated a decrease in the number of hydrogen bonds.

The secondary structure of coconut globulin was mainly composed of α-helix, β-sheet, β-turn, and random coil. The changes in the coconut globulin’s secondary structure are shown in Table 1. The content of the α-helix showed a decreasing trend with the increase in the preheating temperature, which could explain the decrease in solubility after preheating treatment. At high temperatures, the coconut globulin would undergo denaturation, exposing many hydrophobic groups and SH, which could drive the formation of α-helix to decrease solubility. After ultrasound, compared to the preheating treatment alone, some of hydrogen bonds that maintained the α-helix structure were broken, which resulted in a slight decrease in the α-helix content.

#### 3.1.5. Scanning Electron Microscopy of Coconut Protein

Scanning electron microscopy can clearly show the surface characteristics of the protein and its aggregations. As can be seen from Figure 5, the control protein had a non-uniform network with small aggregations, a loose structure and many irregular pores. Compared with the control group, when the protein was preheated at 70 °C, there was an obvious protein aggregation. Moreover, the protein structure became denser, with a more uniform network, larger aggregations and negligible pores. As temperature increased, the protein became a large aggregation, and the irregular pores faded away gradually. This was because the high temperature promoted the expansion of the protein structure. At the same time, the exposure of a large number of hydrophobic groups and the formation of SH stabilized the protein’s structure. In addition, it could be clearly seen that the surface of the aggregation had a large number of granular protrusions. This might be because the unfolding of the coconut globulin side-chain gave itself a more stretched structure. After adding ultrasound to the preheating process, the large protein aggregation dispersed, and the spatial structure of the protein became looser. The high-speed shear force generated by ultrasound could inhibit the formation of large aggregations during the preheating process. At the same time, ultrasound could also decrease the SH content and increase the surface hydrophobicity, which could also restrain the collisions of coconut globulin to form bigger aggregations.

### 3.2. Properties of Coconut Milk

#### 3.2.1. Zeta Potential

Zeta potential is related to the surface charge of particles, and it can affect the aggregation and dispersion of particles [42]. A greater zeta potential could cause the emulsion to have a bigger electrostatic repulsion force, which could enlarge the distance between particles and inhibit the aggregation of particles. Therefore, it was necessary to maintain the emulsion’s stability by increasing the zeta potential [43]. As can been seen from Figure 6, the control group had the lowest zeta potential (−11 mV). After preheating treatment, the zeta potential of all the samples significantly increased. When the preheating temperature was 90 °C, the zeta potential value decreased to −22 mV. For the stable emulsion, such as coconut milk, the negative charge mainly came from the carboxy produced by the dissociation of the side-chain of the coconut globulin amino acid. After preheating, the coconut globulin structure will stretch and expose more carboxy; thus, the zeta potential value would increase with the increase in temperature. It is worth noting that when ultrasound was combined with the preheating process, the energy generated by ultrasound cavitation further stimulated the stretching of the protein’s structure and facilitated the transfer of charge to the surface of the coconut globulin, resulting in a further increase in the zeta potential. Moreover, the further increase in zeta potential could enhance the electrostatic repulsion between the proteins and make oil droplets difficult to accumulate, thereby improving the stability of coconut milk.

It was obvious that the subsequent high-temperature sterilization increased the zeta potential of coconut milk. For example, the zeta potential of the control group increased dramatically by approximately 81%, while the change in the zeta potential value of the preheated sample was only 17–22%. Moreover, the sample with preheating combined with ultrasound had the smallest change. This shows that preheating can significantly improve the coconut milk’s stability during high-temperature sterilization. In addition, the energy generated by ultrasound could further modify the protein’s structure to improve its emulsifying properties and achieve the purpose of stabilizing the emulsion during high-temperature sterilization.

#### 3.2.2. Particle Size and Size Distribution

The particle size and particle size distribution of the emulsion can impact the rates of protein emulsion, flocculates and coalesces [44]. The particle size and particle size distribution can be used to investigate the stability of the emulsion. In Figure 7, we can see that with increase in the preheating temperature, the particle size increased from 4.6 (control) to 7.8 μm (90 °C). For coconut milk, the ability of coconut globulin to reduce the interfacial tension and inhibit the aggregation of oil droplets determined the particle size of the droplets. During preheating, the coconut globulin underwent thermal denaturation, which resulted in a decrease in the solubility and formation of large aggregations to preferentially adsorbing on the interface. Then, due to the steric hindrance of large aggregations, coconut globulin molecules could not cover the oil droplets well and resulted in the coalesce of oil droplets, which ultimately produced a larger particle size. When adding ultrasound to the preheating process, the particle size did not change significantly. However, it was interesting that the particle size of the 90 °C-ultra group produced a significant drop. Under the stimulation of high temperature and ultrasound, the degree of coconut globulin denaturation increased and showed higher surface hydrophobicity, which enhanced the emulsifying properties so that more oil droplets could be fully covered by the coconut globulin, preventing oil droplet aggregation. Therefore, the particle size showed a slight decrease.

When coconut milk was treated by high-temperature sterilization, the particle size of all samples increased. However, it is worth noting that with the increase in preheating temperature, the particle size increased in value during the high-temperature sterilization process gradually becoming smaller. The particle size increase rates of the different samples were 98.3% (70 °C), 77.7% (80 °C), 29.5% (90 °C), 65.5% (70 °C-ultrasound), 39.0% (80 °C-ultrasound) and 13.4% (90 °C-ultrasound), while the control was as high as 148.5%. This was because the coconut globulin molecules would be thermally denatured during the preheating and ultrasound, which resulted in the spatial structure being expanded and a large number of hydrophobic groups being exposed. Thus, the coconut globulin was more easily adsorbed onto the interface, and the oil droplets were better emulsified. Therefore, in the subsequent high-temperature sterilization process, coconut milk could better resist the high-temperature stimulation and maintain its stability.

In Figure 8, it can be seen from the particle size distribution curve that during the process of high-temperature sterilization, the curve of the control group shifted 35.98 μm to the right, and the width of the curve was obviously larger, which shows that the coconut milk particles’ uniformity was reduced and that there was a wider particle size distribution. On the contrary, the right shift distance of the particle size distribution curve for the preheating sample was only 15.36 (70 and 80 °C) and 4.54 μm (90 °C), which was significantly smaller than the control group. This again suggests that preheating can improve the thermal stability of coconut milk. Interestingly, after preheating and ultrasound, the right shift distance and the width of the particle size distribution curve became smaller, among them, the shift distance of the 90 °C-ultra group was only 2.15 μm, which was clearly smaller than for the control group. This also indicates, once again, that preheating and ultrasound treatment could modify the coconut globulin’s structure and improve the emulsifying and interface properties of the protein, thus improving the thermal stability of coconut milk.

#### 3.2.3. Rheological Properties

Rheological property is an inherent physical property of fluid. Moreover, viscosity is used to represent the force of fluid resistance to flow and is related to molecular interactions [45]. The viscosity of coconut milk treated by preheating and ultrasound is shown in Figure 9. It can be seen that as the shear rate increased from 0.01 to 50 s^−1^, the apparent viscosity of all samples decreased and showed the typical shear thinning. It is worth noting that the viscosity of coconut milk gradually increased with the increase in preheating temperature. The viscosity of coconut milk was 262.2 (70 °C), 541.9 (80 °C) and 618.5 m·Pa·s (90 °C), respectively. During the preheating process, the rheological properties of coconut milk mainly came from the contribution of hydrophobic interactions and a small number of disulfide bonds. Under the stimulation of high temperature, the structure of coconut globulin unfolded, the conformation of coconut globulin changed, and a large number of hydrophobic residues were exposed to the environment, which caused the hydrophobic aggregation of the interface proteins to form protein aggregation; then, this aggregation further formed a gel network. At the same time, as the temperature increased, the denaturation degree of coconut globulin increased, which promoted the formation of more protein aggregations. Therefore, the formation of the aggregation resulted in a gradually increase in the apparent viscosity.

Interestingly, when the samples were treated by ultrasound and preheating, the apparent viscosity of the samples showed a trend towards a further increase. The increase rate was 44.0% (70 °C-ultra, 262.2 to 377.6 m·Pa·s), 80.7% (80 °C-ultra, 541.9 to 979.1 m·Pa·s) and 78.3% (90 °C-ultra, 618.5 to 1103.0 m·Pa·s). This was because ultrasound can promote the formation of some tiny fat crystals, which could lead to droplet coalescence. Ultrasound can promote the formation of fat crystal bridges between droplets and induce the dissociation of protein subunits to improve the viscosity of coconut milk [46]. Moreover, the increase in the surface hydrophobicity and the decrease in the SH content, caused by ultrasound, cause coconut globulin aggregations to combine with the coconut globulin adsorbed on the oil–water interface, which increases the apparent viscosity.

## 4. Conclusions

In summary, ultrasound combined with preheating technology provided coconut globulin with excellent physicochemical properties. The higher solubility, fewer SH and increased surface hydrophobicity improved the ability of coconut globulin to emulsify oil and made coconut globulin more resistant to heat damage and able to maintain its structural stability. The results also showed that ultrasound combined with preheating induced changes in the secondary structure of coconut globulin with an increase in α-helix, β-sheet and random coil and a decrease in β-turn, suggesting a transition in the coconut globulin conformation from disorder to order and an increase in the structure’s stability. Moreover, the increase in the zeta potential and viscosity and the decrease in the thermal aggregation rate of coconut milk, caused by the combination of ultrasound with preheating, showed that coconut milk had better thermal stability. Overall, this study suggests that ultrasound combined with preheating can be used to improve the physicochemical properties of coconut globulin and increase the thermal stability of coconut milk. In the future, this technology may be applied to more protein modification studies, and the mechanism of action will be further explored. The improvement in the coconut milk’s stability can be applied to solve the problems encountered in the processing of coconut milk. It will reduce the use of additives, decrease production costs and ensure the green and nature of the products. At the same time, the application of this technology can also promote the development of relative coconut industries and improve the economy of tropical regions.

## Figures and Tables

**Figure 1 foods-11-01042-f001:**
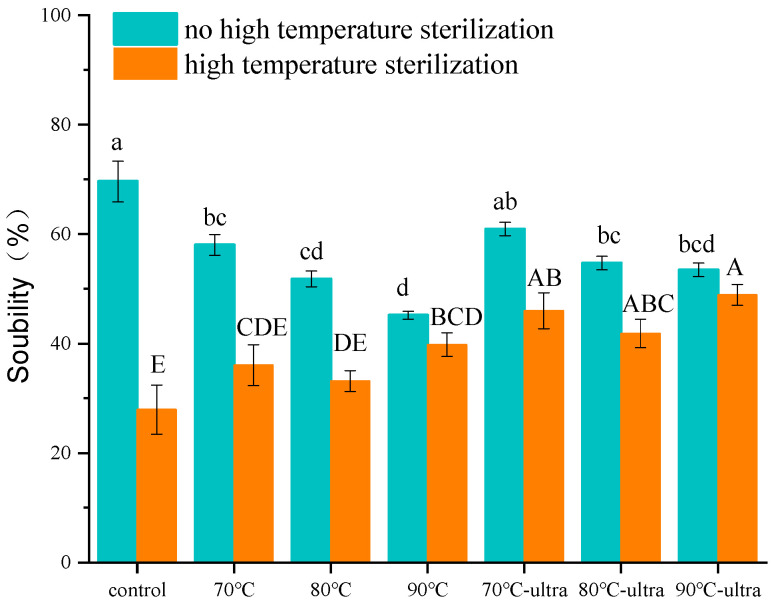
Solubility of different coconut globulin samples (ultra: 53 kHz, 40 *w*/*l*; 70, 80 and 90 °C represent preheating for 20 min; high-temperature sterilization: 135 °C for 60 s). Different letters indicate significant differences at *p* < 0.05.

**Figure 2 foods-11-01042-f002:**
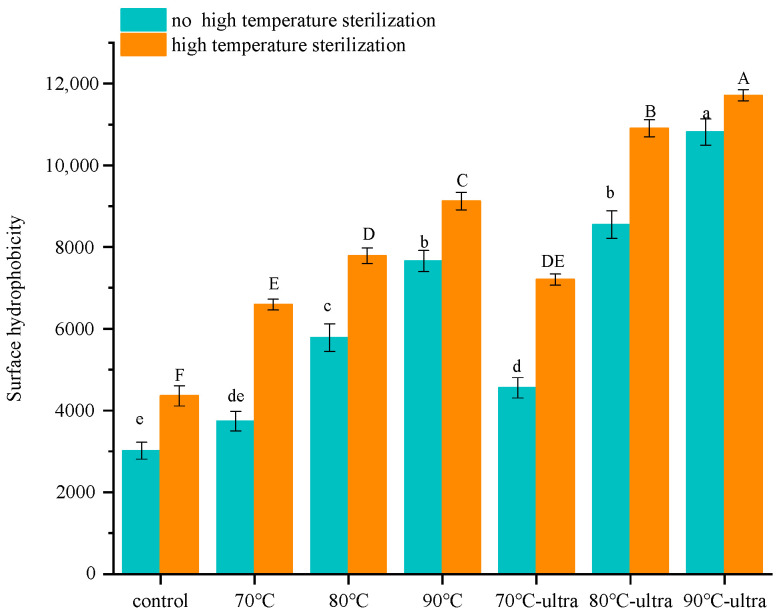
Surface hydrophobicity of different coconut globulin samples (ultra: 53 kHz, 40 *w*/*l*; 70, 80 and 90 °C represent preheating for 20 min; high-temperature sterilization: 135 °C for 60 s). Different letters indicate significant differences at *p* < 0.05.

**Figure 3 foods-11-01042-f003:**
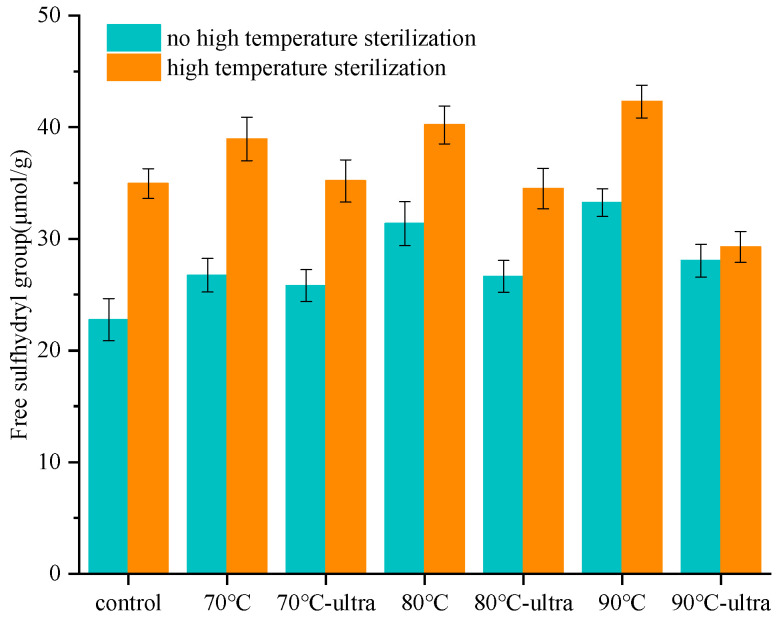
SH of different coconut globulin samples (ultra: 53 kHz, 40 *w*/*l*; 70, 80 and 90 °C represent preheating for 20 min; high-temperature sterilization: 135 °C for 60 s).

**Figure 4 foods-11-01042-f004:**
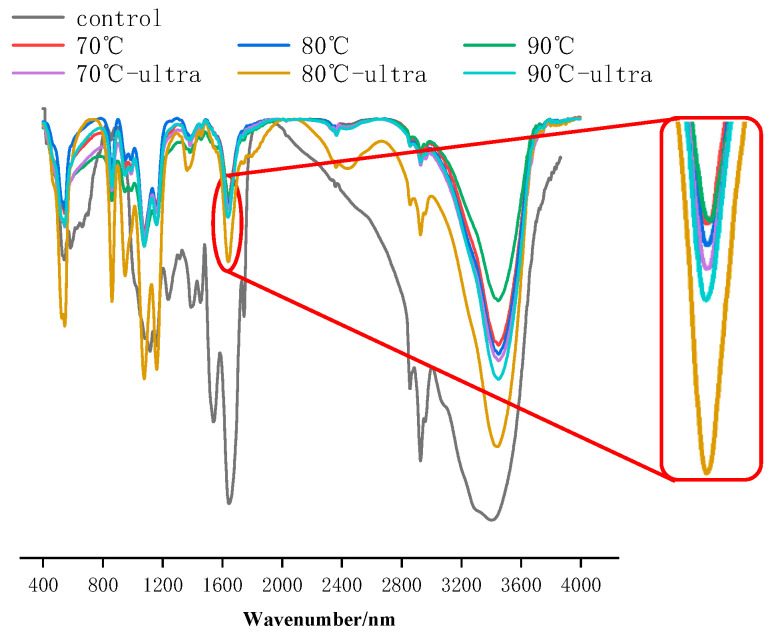
FTIR of different coconut globulin samples (ultra: 53 kHz, 40 *w*/*l*, 20 min; 70, 80 and 90 °C represent preheating for 20 min).

**Figure 5 foods-11-01042-f005:**
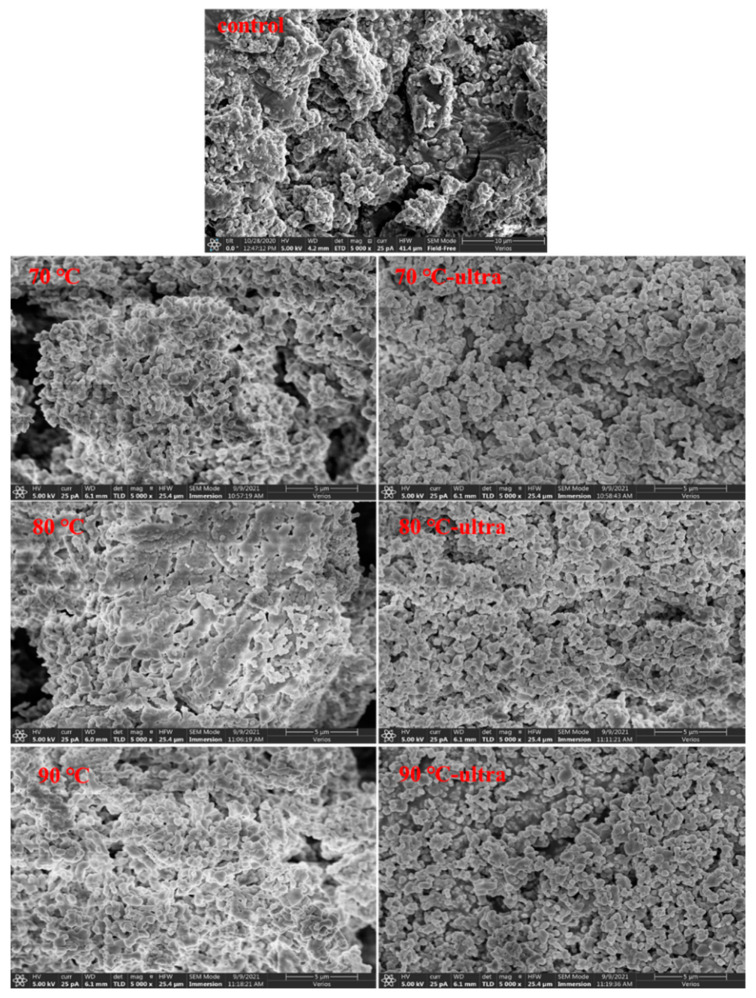
Scanning electron microscopy of different coconut globulin samples (ultra: 53 kHz, 40 *w*/*l*, 20 min; 70, 80 and 90 °C represent preheating for 20 min).

**Figure 6 foods-11-01042-f006:**
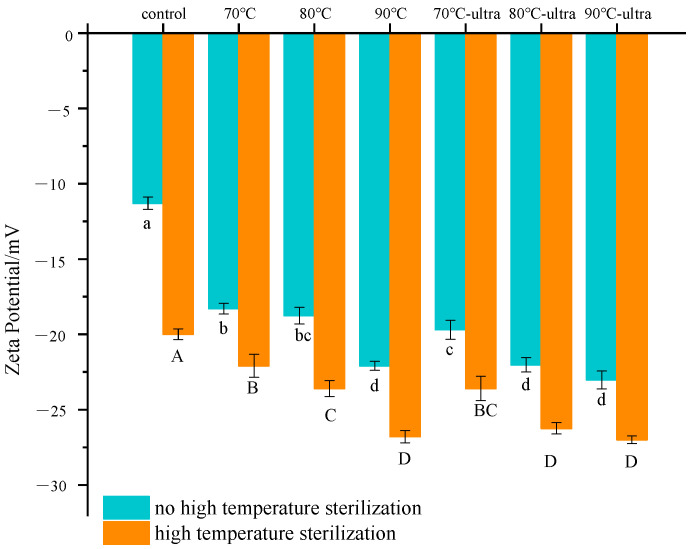
Zeta potential of different coconut milk samples (ultra: 53 kHz, 40 *w*/*l*; 70, 80 and 90 °C represent preheating for 20 min; high-temperature sterilization: 135 °C for 60 s). Different letters indicate significant differences at *p* < 0.05.

**Figure 7 foods-11-01042-f007:**
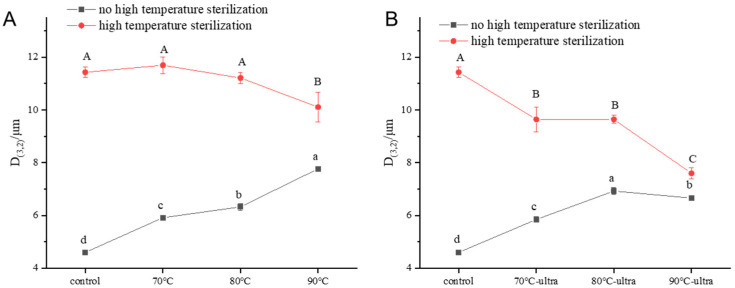
Particle size of different coconut milk samples ((**A**): no ultrasound; (**B**): ultrasound, 53 kHz, 40 *w*/*l*, 20 min; 70, 80 and 90 °C represent preheating for 20 min; high-temperature sterilization: 135 °C for 60 s). Different letters indicate significant differences at *p* < 0.05.

**Figure 8 foods-11-01042-f008:**
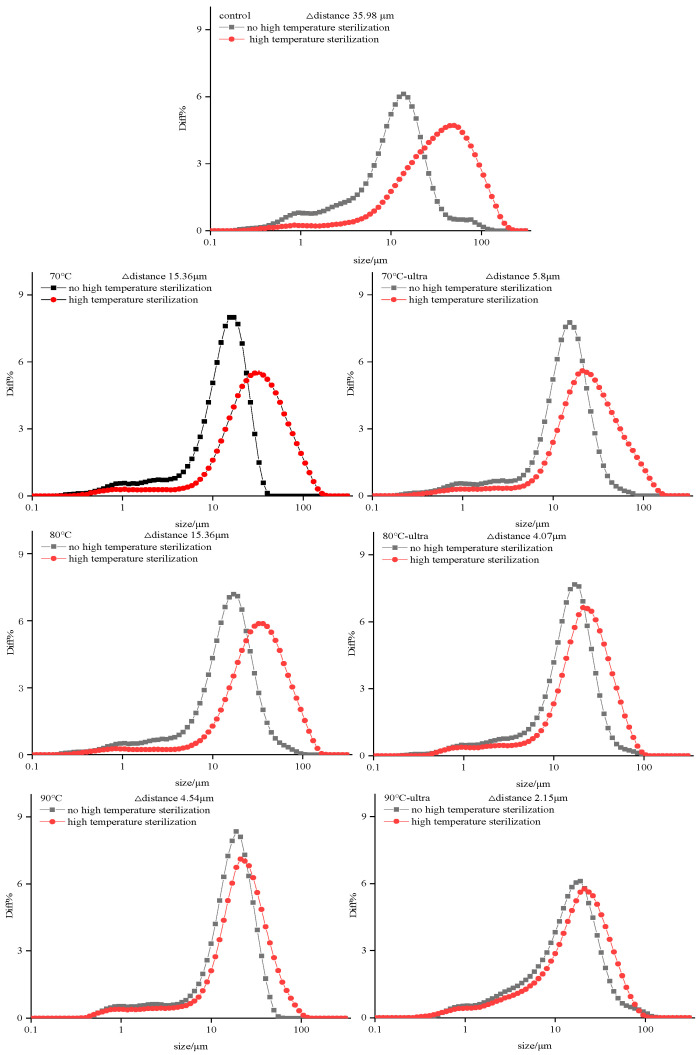
Particle size distribution curve of different coconut milk samples before and after thermal treatment (ultra: 53 kHz, 40 *w*/*l*, 20 min; 70, 80 and 90 °C represent preheating for 20 min; high-temperature sterilization: 135 °C for 60 s).

**Figure 9 foods-11-01042-f009:**
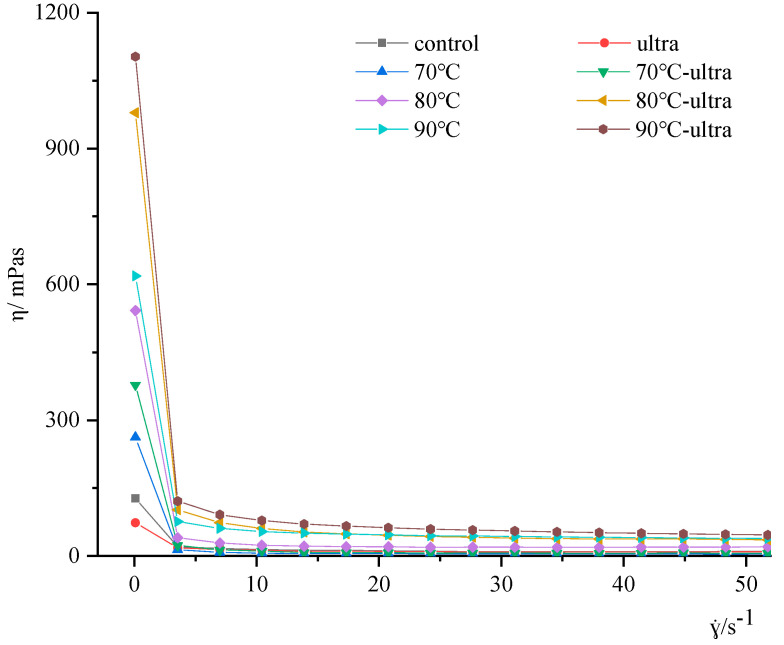
Rheological properties of different coconut milk samples before and after thermal (ultra: 53 kHz, 40 *w*/*l*, 20 min; 70, 80 and 90 °C represent preheating for 20 min).

**Table 1 foods-11-01042-t001:** Secondary structures of different coconut globulin samples (ultra: 53 kHz, 40 *w*/*l*, 20 min; 70, 80 and 90 °C represent preheating for 20 min). Different letters indicate significant differences at *p* < 0.05.

Sample	α-Helix (%)	β-Sheet (%)	β-Turn (%)	Random Coil (%)
control	20.82 ± 0.45 ^e^	22.13 ± 0.56 ^e^	35.74 ± 1.31 ^a^	21.31 ± 0.40 ^b^
70 °C	21.94 ± 0.07 ^abc^	32.21 ± 0.13 ^b^	19.99 ± 0.03 ^cd^	25.86 ± 0.07 ^a^
80 °C	22.31 ± 0.07 ^ab^	30.56 ± 0.67 ^c^	21.16 ± 0.01 ^bc^	25.98 ± 0.78 ^a^
90 °C	22.65 ± 0.06 ^a^	29.12 ± 0.12 ^d^	22.96 ± 0.02 ^b^	25.29 ± 0.08 ^a^
70 °C-ultra	21.43 ± 0.07 ^cd^	32.26 ± 0.14 ^b^	20.85 ± 0.30 ^cd^	25.47 ± 0.09 ^a^
80 °C-ultra	21.64 ± 0.08 ^bcd^	32.60 ± 0.16 ^ab^	19.62 ± 0.02 ^cd^	26.15 ± 0.10 ^a^
90 °C-ultra	21.53 ± 0.08 ^bcd^	33.74 ± 0.15 ^a^	18.97 ± 0.02 ^d^	25.78 ± 0.09 ^a^

## Data Availability

Not applicable.

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
