# Peer review of "Effects of Ultrasound Combined with Preheating Treatment to Improve the Thermal Stability of Coconut Milk by Modifying the Physicochemical Properties of Coconut Protein"

_foods, 2022, doi:10.3390/foods11071042_

Round 1

Reviewer 1 Report

This is quiet an interesting and applicative study from industrial point of view. It is a study which is well designed. 

Please add more information in the introduction section egarding significance of these kind of research from industrial point of view and applicability. Please add info how much cocconut milk is used per annum globaly. 

Lines 62-65 you need references for these statements

Material and methods section are clear.

Results are very well presented and it seems clear with appropriate discussion. 

Please add, in conclusion section, future perspectives of this work and where do you expect application of these findings and what are the benefits.

Author Response

Reviewer #1: This is quiet an interesting and applicative study from industrial point of view. It is a study which is well designed.

Thanks for the positive feedback.

Reviewer #1: Please add more information in the introduction section egarding significance of these kind of research from industrial point of view and applicability. Please add info how much cocconut milk is used per annum globaly.

“Coconut milk industry plays an important role in economic development for tropical areas. Coconut milk is high in fat, stabilized by natural coconut protein, and has excellent nutritional value. Due to the bad emulsifying properties of coconut protein, coconut milk is prone to be unstable and stratified in the process of production, transportation and processing, which affects the product quality. Because, it is important to maintain the stability of coconut milk in industry. The study of improving protein emulsifying properties by preheating and ultrasound can reduce the use of additives, ensure the safety, nature, and healthy of products. At the same time, it is beneficial to reduce the production cost, optimize the production process and promote the development of coconut industry.” We have added this content in Line 92-100.

“Study indicated that the global coconut milk market was estimated at $814.4 million in 2020 and the coconut milk market will grow at 9.3% Compound Annual Growth Rate to reach $1,775.4 million by 2027.” We have added this content in Line 32-34.

Reviewer #1: Lines 62-65 you need references for these statements

We had added some references in Line 91 and 104.

Reviewer #1: Material and methods section are clear.

Thanks for the comment.

Reviewer #1: Results are very well presented and it seems clear with appropriate discussion.

Thank the reviewer for the positive feedback.

Reviewer #1: Please add, in conclusion section, future perspectives of this work and where do you expect application of these findings and what are the benefits.

“In the future, this technology may be applied in more protein modification studies and the mechanism of action will be further explored. The improvement of coconut milk stability can be applied to solve the problems encountering in the processing of coconut milk. It will reduce the use of additives, decrease production costs, and ensure green and nature of products. At the same time, the application of this technology can also promote the development of relative coconut industries, and improve the economy of tropical regions.” We had added some references in Line 572-578.

Reviewer 2 Report

The presentation of the results in the Abstract chapter is too general. Present some concrete results, data.

The presentation of the results is too descriptive. There is very short comparison of the results of the experiment with studies by other authors or even with other products with a similar role (e.g. milk).

Comparing the parameters of coconut milk as a product of plant origin with the parameters of milk of animal origin would be very interesting. It would be similarly interesting to compare coconut milk with other plant products.   

Author Response

Reviewer #2: The presentation of the results in the Abstract chapter is too general. Present some concrete results, data.

We have refined the abstract by adding concrete results and data in Line 18-23.

Reviewer #2: The presentation of the results is too descriptive. There is very short comparison of the results of the experiment with studies by other authors or even with other products with a similar role (e.g. milk).

In our previous studies, we found that ultrasound can extend the storage life of fresh coconut water, improve the functional properties of coconut protein and increase the stability of coconut protein emulsion. These results are similar to some other studies with soy protein. Additions, as a convenient and green physical modification technology, preheating may have a broad development prospect in protein modification and emulsion application. Therefore, we research the technology of ultrasound combined with preheating in this study. The results showed that the technology of combining ultrasound and preheating could better improve the physicochemical properties of coconut protein to increase the stability of coconut milk. However, the similar studies are rarely reported on this technology and the technology is not applied on other proteins. Therefore, we haven't made any similar comparisons.

Reviewer #2: Comparing the parameters of coconut milk as a product of plant origin with the parameters of milk of animal origin would be very interesting. It would be similarly interesting to compare coconut milk with other plant products.

“Study indicated that coconut milk is a good substitute for cow milk. On the one hand, cow milk contains a lot of lactose which has the highest allergenic potential, while coconut doesn’t contain lactose. On the other hand, the fat content of cow milk is 3.27 g/100ml and coconut milk is 5.04 g/100ml. But, about 87% fat of coconut milk is lauric acid, followed by caprylic acid and decanoic acid (13%). This fat is more easily metabolized by body and the energy is only 70 kcal/100ml, which is lower than the cow milk (150kcal kcal/100ml). At the same time, this fat is beneficial to prevent arteriosclerosis and related diseases. Compared with soy milk, the protein content of coconut milk (1.28 g/100ml) is lower than soy milk (2.88 g/100ml). However, soy protein is considered an allergen and soy milk carries a higher risk of allergy. Coconut protein doesn’t have the risk of allergy and contains a large number of essential amino acids (71%-77%), which is mor easily to be digested and absorbed by body. Moreover, compared with soy milk, coconut milk contains more minerals (particularly calcium, phosphorus, potassium) and vitamins (vitamins C, E, B1, B3, B5, B6), which has strong antioxidant activity [3] Therefore. coconut milk is an excellent vegetable protein drink” We had added this content in Line 34-61.
